# Non-Targeted Detection and Quantification of Food Adulteration of High-Quality Stingless Bee Honey (SBH) via a Portable LED-Based Fluorescence Spectroscopy

**DOI:** 10.3390/foods12163067

**Published:** 2023-08-15

**Authors:** Diding Suhandy, Dimas Firmanda Al Riza, Meinilwita Yulia, Kusumiyati Kusumiyati

**Affiliations:** 1Department of Agricultural Engineering, Faculty of Agriculture, The University of Lampung, Jl. Soemantri Brojonegoro No. 1, Bandar Lampung 35145, Indonesia; 2Department of Biosystems Engineering, Faculty of Agricultural Technology, University of Brawijaya, Jl. Veteran, Malang 65145, Indonesia; dimasfirmanda@ub.ac.id; 3Department of Agricultural Technology, Lampung State Polytechnic, Jl. Soekarno Hatta No. 10, Bandar Lampung 35141, Indonesia; meinilwitayulia@polinela.ac.id; 4Department of Agronomy, Faculty of Agriculture, Universitas Padjadjaran, Sumedang 45363, Indonesia; kusumiyati@unpad.ac.id

**Keywords:** authentication, food adulteration, *Heterotrigona itama*, multivariate calibration, portable LED-based fluorescence spectroscopy, stingless bee honey (SBH)

## Abstract

Stingless bee honey (SBH) is rich in phenolic compounds and available in limited quantities. Authentication of SBH is important to protect SBH from adulteration and retain the reputation and sustainability of SBH production. In this research, we use portable LED-based fluorescence spectroscopy to generate and measure the fluorescence intensity of pure SBH and adulterated samples. The spectrometer is equipped with four UV-LED lamps (peaking at 365 nm) as an excitation source. *Heterotrigona itama*, a popular SBH, was used as a sample. 100 samples of pure SBH and 240 samples of adulterated SBH (levels of adulteration ranging from 10 to 60%) were prepared. Fluorescence spectral acquisition was measured for both the pure and adulterated SBH samples. Principal component analysis (PCA) demonstrated that a clear separation between the pure and adulterated SBH samples could be established from the first two principal components (PCs). A supervised classification based on soft independent modeling of class analogy (SIMCA) achieved an excellent classification result with 100% accuracy, sensitivity, specificity, and precision. Principal component regression (PCR) was superior to partial least squares regression (PLSR) and multiple linear regression (MLR) methods, with a coefficient of determination in prediction (R^2^_p_) = 0.9627, root mean squared error of prediction (RMSEP) = 4.1579%, ratio prediction to deviation (RPD) = 5.36, and range error ratio (RER) = 14.81. The LOD and LOQ obtained were higher compared to several previous studies. However, most predicted samples were very close to the regression line, which indicates that the developed PLSR, PCR, and MLR models could be used to detect HFCS adulteration of pure SBH samples. These results showed the proposed portable LED-based fluorescence spectroscopy has a high potential to detect and quantify food adulteration in SBH, with the additional advantages of being an accurate, affordable, and fast measurement with minimum sample preparation.

## 1. Introduction

Two types of honey are now available in Indonesia that are premium-priced: floral honey (FH) (*Apis mellifera*) and stingless bee honey (SBH) [1]. The entomological morphology of the two honeybees is different. *Apis mellifera* (FH) is bigger in size, has a sting, and produces sweet honey, while SBH is smaller in size, stingless, and produces a mixture of sweet and sour-tasting honey [2,3]. One of the distinguishing features of honey and its derivative products (propolis, royal jelly, and bee pollen) is that they are high in bioactive compounds (total phenolic compounds and flavonoid contents) and are widely used in the food, cosmetic, and pharmaceutical industries [4,5]. Of particular interest, honey is a natural food that is proven to increase human immunity. Honey is effective against virus-based diseases (such as influenza and HIV viruses) and thus potentially effective for COVID-19 [6,7,8]. In particular, SBH contains more total phenolic compounds and flavonoid contents and produces superior antioxidant properties compared to FH [9,10,11]. In Indonesia, three popular and profitable SBHs are *Heterotrigona itama*, *Tetrigona apicalis,* and *Geniotrigona thoracica*. SBH is more expensive compared to FH (non-SBH) due to its limited production and high flavonoids and polyphenolic content, thus making it a target of economically motivated adulteration [EMA] [12]. In the market, SBH is often adulterated with low-cost honey or cheaper artificial industrial sweeteners, such as HFCS-55 (high fructose corn syrup), glucose syrup, rice syrup, and invert sugar syrups, resulting in inferior nutrient content as well as potential food safety hazards in the final commercial SBH [13,14,15,16].

The sugar composition of SBH depends mainly on the source of nectar [17]. Consequently, the physicochemical analysis of individual SBH samples for honey authenticity tests is unreliable. According to Codex Alimentarius (Codex Alimentarius, 2001), the adulteration of honey with sugar cane or corn syrups, including HFCS, is currently verified by stable carbon isotope ratio analysis (SCIRA) or by the sugar profile determined by high-performance liquid chromatography (HPLC). However, these two standard methods include costly instrumentation, complex techniques, laborious sample preparation, and time-consuming analysis [18].

More recent honey authenticity detection techniques, including near-infrared (NIR), Raman, and fluorescence spectroscopy [19,20,21,22,23,24,25,26], also suffer from several limitations, such as being time-consuming, involving expensive benchtop-based instruments, requiring a highly-trained operator to perform the analysis, and in a number of cases generating toxic waste and using expensive chemicals. On the other hand, the use of easy, fast, and chemical-free analytical methods based on ultraviolet-visible (UV-visible) spectroscopy with a benchtop spectrometer for testing the authenticity of food has been well reported in Indonesia, especially for coffee [27,28,29], tea [30], and honey [31].

Recently, the development of compact and portable spectrometers has increased and is now available in UV-visible to near-infrared regions and Raman spectroscopy [32,33,34]. It is a promising technology for rapid, low-cost, and on-site honey analysis, including botanical and geographical detection [35], honey authentication [36], and traceability evaluation [37]. This portable spectroscopy combined with appropriate chemometrics offers several advantages compared to lab/benchtop-based spectroscopy, such as fast spectral acquisition and analysis time, affordable operating and maintenance costs, low energy consumption, easy operation, and affordability for field work in developing countries, with Indonesia being no exception [38]. Portable spectrometers can be implemented in situ, allowing the flexibility of on-site measurements, minimizing sample transport errors, and providing real-time responses [39].

To develop a compact and cost-effective portable spectrometer, the use of a light-emitting diode (LED) as the light source for excitation is also promising. LED is a semiconductor source that offers extremely low power consumption and operating voltage, compactness, and durability, and a high cost/performance ratio compared to other light sources [40]. On the other hand, there are several drawbacks to LED-based fluorescence spectroscopy, such as its limited spectral range and the necessity of arranging multiple LEDs when multiple excitation wavelengths are required [41,42]. This results in a more complex system arrangement compared to the system that utilizes polychromatic light sources. LED-based fluorescence spectroscopy with multiple LED systems will also have a low spectral resolution compared to conventional spectroscopy systems [43]. To date, though, limited studies have explored the use of LED-based fluorescence spectroscopy equipped with a single LED for honey authentication [44,45]. For this reason, in this study, we evaluate the use of a compact, inexpensive, and portable fluorescence spectrometer equipped with four single UV-LED lamps (peak at λ = 365 nm) as excitation sources to generate and then measure the intensity of fluorescent components in pure SBH samples as well as SBH samples that have been adulterated. In this study, HFCS-55 (high fructose corn syrup), an inexpensive and commonly used artificial sweetener in the food industry, was used as the adulterant. Chemometric models were developed for identification (PCA and SIMCA) and quantification (PLSR, PCR, and MLR) of corn syrup adulterants in SBH samples.

## 2. Materials and Methods

### 2.1. Stingless Bee Honey (SBH) Samples

The stingless bee honey (SBH) of monofloral *Heterotrigona itama* (floral nectar *Acacia mangium*) was collected and processed in February 2023 by a registered and reputable stingless bee honey processing farm, PT Suhita Lebah Indonesia, located in Bandar Lampung, Lampung, Sumatra, Indonesia, with known GPS locations (5°27′ S and 105°16′ E latitude and longitude, respectively, with 100 m altitudes) (Figure 1). The samples were transported to the Bioprocess and Postharvest Engineering Laboratory at The University of Lampung and stored at about 15 °C in the dark to avoid direct light exposure before fluorescence spectral data was acquired. The adulterated samples were generated by directly mixing pure SBH with varying amounts of high fructose corn syrup (HFCS-55): from 10–60% (volume/volume) adulteration.

In this study, we measured three hundred forty samples of pure and adulterated SBH (n = 100 for pure *Heterotrigona itama* and *n* = 240 for adulterated *Heterotrigona itama*, respectively). More detailed information on the pure and adulterated SBH samples used is described in Table 1. The color of pure SBH (0% adulteration or MA) samples was darker compared to their adulterated ones (Figure 2). It was reported that honey color presents variations depending on the contents of beta-carotene, chlorophyll and its derivatives, and anthocyanins, and thus there is a correlation between color and the total phenolic content and antioxidant activity. As a result, the color of honey highly depends on its botanical source [46,47,48]. For this reason, the discrimination between the pure and adulterated SBH samples based solely on color features, especially at low adulteration levels, is not reliable.

### 2.2. Sample Preparation

Sample preparation of pure and adulterated SBH samples was done using a simple protocol (Figure 3). All samples were heated in a water bath at 60 °C for 30 min [49]. A previous report noted that dilution significantly affected the intensity of the fluorescence spectra of the diluted samples compared to the pure honey samples [50]. In this study, the pure and adulterated SBH samples were subjected to dilution with distilled water (at 1:5 volume/volume proportion) to obtain a diluted honey sample [50]. The diluted honey samples were stirred well using a magnetic stirrer (CiBlanc) for 10 min before fluorescence spectral data acquisition.

### 2.3. Fluorescence Spectral Data Acquisition

A portable fluorescence spectrometer from GoyaLab (IndiGo Fluo UV spectrometer, Talence, France) working with 4 LED lamps with an excitation wavelength of λ = 365 nm and 0.5 nm of resolution was connected to a personal computer. This spectrometer was utilized to obtain fluorescence emission spectra of pure and adulterated SBH samples over the range of 357 to 725.5 nm (Figure 3). For each sample, two mL of diluted sample were pipetted into a quartz cuvette with a 10 mm path length. The spectral acquisition was performed using 2000 ms of exposure time, and it was controlled by the SpectroLab application installed on the personal computer free of charge. The original fluorescence spectral data of all the SBH samples (*Heterotrigona itama*) were transferred to a computer for further chemometric analysis. A moving average smoothing algorithm with 15 segments of smoothing points was used to smooth and increase the signal-to-noise ratio (SNR) of the obtained original fluorescence spectral data.

### 2.4. Chemometric Analysis

Chemometrics can be defined as “the chemical discipline that uses mathematical, statistical and other methods employing formal logic to design or select optimal measurement procedures and experiments, and to provide maximum relevant chemical information by analysing chemical data” [51]. Fluorescence spectroscopy generates rich-overlapped spectral data, and it is hard to understand the phenomenon directly from the raw spectral data. From this point, chemometrics are useful techniques to separate informative data from noise, extract hidden correlations, and provide a visual approach for both qualitative and quantitative multivariate data analysis. In general, chemometric approaches are applied for explorative analysis, classification, and multivariate calibration purposes [52]. In this study, the explorative analysis and classification between pure and adulterated SBH samples are evaluated by using principal component analysis (PCA) and soft independent modeling of class analogy (SIMCA) methods, respectively. There are several algorithms to determine the optimal principal components (PCs), including singular value decomposition (SVD) and nonlinear iterative partial least squares (NIPALS) algorithms. In this study, PCA was calculated for all pure and adulterated samples (n = 340) with the NIPALS algorithm, a common algorithm for the analysis of complex data, to generate the score and loading of each PC. The score plot of the first two PCs (PC1 × PC2) was utilized to evaluate the possible separation between pure and adulterated SBH samples, while the plot of x-loadings for PC1 and PC2 was used to identify the important variables responsible for the separation. The theory and detailed explanation of PCA have been discussed in several studies [53,54].

A well-known supervised classification was developed by the SIMCA method using three sample sets (Table 2). The SIMCA model for each class of pure and adulterated SBH was developed using a calibration sample set for each class (n = 51 and n = 126 for pure and adulterated SBH samples, respectively) and then validated using a *t*-test validation method (n = 33 and n = 78 for pure and adulterated SBH samples, respectively). The prediction sample set (unknown samples) (n = 16 and n = 36 for pure and adulterated SBH samples, respectively) was then compared to the class models and its members assigned to classes according to two criteria: the distance from the model center (leverage) and the distance to the model (residual). A confusion matrix was generated to visualize the classification result of SIMCA. The sensitivity, specificity, accuracy, and precision were used to assess the results of the classifications and were calculated according to Equations (1)–(4), respectively [55,56]. In this study, pure SBH was denoted as “positive” and adulterated SBH as “negative”. The terms FP, FN, TP, and TN mean the numbers of false positives, false negatives, true positives, and true negatives, respectively [56]. The classification results of SIMCA are shown at a significance level of 0.05 (α = 0.05).
(1)$$\mathrm{Sensitivity}=\frac{\mathrm{TP}}{\mathrm{TP}+\mathrm{FN}}$$
(2)$$\mathrm{Specificity}=\frac{\mathrm{TN}}{\mathrm{TN}+\mathrm{FP}}$$
(3)$$\mathrm{Accuracy}=\frac{\mathrm{TN}+\mathrm{TP}}{\mathrm{TN}+\mathrm{TP}+\mathrm{FN}+\mathrm{FP}}$$
(4)$$\mathrm{Precision}=\frac{\mathrm{TP}}{\mathrm{TP}+\mathrm{FP}}$$

To quantify the pure and adulterated SBH samples, three regression methods based on partial least squares regression (PLSR), principal component regression (PCR), and multiple linear regression (MLR) were used as statistical methods to develop a model for training with fluorescence spectra of pure and adulterated SBH samples having known adulteration levels. The developed model predicts the unknown adulteration level based on the captured fluorescence spectra. PLSR utilizes all fluorescence spectra data as predictors and aims to develop a linear regression model using a latent variable (LV) approach by projecting the predicted variables and the observable variables to a new space. PCR aims to find hyperplanes of maximum variance between the two variables. PLSR and PCR were developed using fluorescence spectra from 357 to 725.5 nm (number of variables = 738). In MLR, several selected wavelengths were used as predictors to build a linear relationship between the fluorescence spectral data and adulteration level. PLSR, PCR, and MLR models were validated by a *t*-test validation method to optimize the model parameters. In accordance with Yulia and Suhandy [29], several statistical parameters were used to assess the calibration model, including the coefficients of determination of calibration and validation (R^2^_c_ and R^2^_v_), root means squared errors of calibration and validation (RMSEC and RMSEV), the ratio of prediction to deviation in prediction (RPD), and the range error ratio (RER) index. The limit of detection (LOD) and limit of quantification (LOQ) were also calculated according to Milani et al. [57] and Rambla-Alegre et al. [58], respectively. SD is the residual standard deviation of the regression curve (or standard error of prediction (SEP)), and S is the slope [58]. For quantification purposes, the pure and adulterated SBH samples were divided randomly into three different sample subsets: calibration, validation, and prediction data sets (Table 3). All chemometric calculations were conducted using the Unscrambler^®^, version 10.5 from CAMO (Norway).
(5)$$\mathrm{LOD}=\frac{3\times \mathrm{SD}}{S}$$
(6)$$\mathrm{LOQ}=\frac{10\times \mathrm{SD}}{S}$$

## 3. Results and Discussion

### 3.1. Fluorescence Spectral Intensity of Pure and Adulterated SBH Samples

Figure 4 shows the smoothed fluorescence spectra of pure and adulterated SBH samples excited using 4 UV-LED lamps (λ = 365 nm). It can be seen that some regions of fluorescence emission could distinguish the different qualities of SBH, especially in the wavelength range of 350–410 nm and 430–570 nm. However, exploration of other excitation wavelengths could be worthwhile. Regarding the intensity, it can be seen that the emission spectra produced in this experiment have a high signal-to-noise ratio (SNR), showing that the intensity of the present excitation light source with four single LED lamps is sufficient. In general, the shape and intensity of the obtained fluorescence spectra were similar to those reported previously [59,60,61]. Two fluorescent peaks were observed at 378 nm and 477 nm. It is observed that the original fluorescence spectra of pure SBH samples were lower compared to the adulterated samples at around 378 nm and 477 nm peaks.

At the 378 nm peak, the fluorescence intensity of adulterated SBH samples increased as the adulteration level increased. In contrast, at the 477 nm peak, the fluorescence intensity of adulterated SBH samples decreased as the adulteration level increased. Our findings are in close agreement with previous work. Ghosh et al. [59] reported the fluorescence of *Apis florea* honey adulterated with cane sugar syrup, a C-4 plant sweetener. In this study, the fluorescence intensity of adulterated honey increased at 365 nm and decreased at 460 nm as the concentration of cane sugar syrup increased [59]. According to previous works, the observed peaks in the current study at 378 and 477 nm are closely related to nicotinamide adenine dinucleotide (NADH) and flavin absorption peaks, respectively [59,60]. The fluorescence intensity (I) ratio between 378 and 477 nm (I_378_/I_477_) could be an important index for characterizing and differentiating between pure and adulterated SBH samples. Nikolova et al. [61] investigated the fluorescence spectra of 24 different types of honey, including *Robinia pseudoacacia* and *Helianthus annuus,* from different botanical and geographical origins in Bulgaria using LED lamps with four excitation wavelengths at 375, 395, 425, and 450 nm. For most samples measured, the peak fluorescence intensity was observed at wavelengths of 490 and 505 nm. Ruoff et al. [62] reported a similar result. With excitation at 290 nm, peak fluorescence of chestnut and honeydew honey was observed at 375 and 410 nm.

### 3.2. PCA and SIMCA Results

Figure 5 shows the result of the PCA calculation. Using the first two PCs that explained 95% of the variance, a separation between pure and adulterated SBH samples was established, especially along the PC1 axis. The addition of HFCS-55 to pure *Heterotrigona itama* honey shifted the samples to the right of PC1 (PC1 scores became more positive). For this, we can see that all pure SBH samples were clustered on the left of PC1 (PC1 score < 0), and most of the adulterated SBH samples were located on the right of PC1 (PC1 score > 0).

The X-loading spectra of the first three PCs were plotted against wavelengths to evaluate the most contributive variables (Figure 6). The positive loading spectra for PC1 showed that the peak at 391 nm has a significant positive contribution, while other peaks at 384 and 470 nm in PC2 and PC3 had a similar impact in the opposite direction. Two positive peaks at 490 and 540 nm were also observed for PC2 and PC3. These important peaks in the fluorescence spectra of honey are consistent with those reported in previous works. For example, the emission at 391 nm is closely related to phenolic compounds, and the emissions at 470 and 490 nm may well be related to caffeic acid, chlorogenic acid, and ferulic acid in honey [63]. The presence of a peak at 391 nm with a positive direction could be understood to be due to the fact that SBH honey is rich in phenolic compounds, as reported in many studies [63]. The intensity of these phenolic compounds is strongly correlated with the antioxidant activity, color, and sensory features of SBH, and it could provide a valuable marker for differentiation between pure and adulterated SBH [64]. The peaks at 470 and 490 nm obtained in our SBH samples have also been observed in several previous studies. Lang et al. [65] reported a fluorescence peak of phenolic substances, such as chlorogenic acid, caffeic acid, and coumarins, close to 450 nm. The emission band for caffeic acid is in the range between 445 and 460 nm, which corresponds to the chlorogenic and ferulic acids of SBH, as reported by Sergiel et al. [66]. For these reasons, the important emission peaks at 391, 384, 470, 490, and 540 nm were selected as inputs for MLR.

Table 4 shows the result of SIMCA model development for pure and adulterated SBH classes. The first six principal components (PCs) were obtained both in calibration and validation for both classes. The cumulative percent variance (CPV) of the first four PCs was more than 99% for calibration and validation, both for pure and adulterated SBH classes. It should be noted that a CPV of more than 70–85% is required to establish a reliable SIMCA model [67]. Thus, the developed SIMCA model of pure and adulterated SBH classes was used to predict the class of new SBH samples in the prediction sample set (*n* = 52 for pure and adulterated SBH samples). The result is presented in Table 5.

The pure and adulterated prediction samples were classified with 100% accuracy, sensitivity, specificity, and precision, resulting in excellent classification. To confirm the result of classification, a model distance between the pure and adulterated SBH classes was calculated. A distance larger than 3 indicates good class separation and a low risk of misclassification in the model. A model distance of 121.6199 was obtained in this study (much larger than 3). This indicates the pure and adulterated SBH models were highly differentiating, providing a low risk of misclassification in the model. The plot of model distance (Si) versus leverage (Hi) for the pure SBH model is shown in Figure 7, demonstrating that there is no overlap between any of the pure and adulterated samples (all adulteration levels), with no sample classified as false positive or false negative (FP and FN equal to 0). According to Se et al. [12], the limit of detection (LOD) of the SIMCA model can be calculated based on only the true positive (TP) results. Consequently, the developed SIMCA model with smoothed fluorescence spectra had excellent classification for the adulterated SBH samples at concentrations above 10% (*v*/*v*). This LOD value for the developed SIMCA model is consistent with the result of a previous study. For example, Se et al. [12] proposed the rapid detection and quantification of adulterants in stingless bee honey (*Heterotrigona itama*) using an FTIR approach. The LOD was 2% (*w*/*w*) and 8% (*w*/*w*) for sugar cane and corn syrup adulteration, respectively. Our result was also comparable with the standard analysis technique for honey adulteration detection based on stable carbon isotope ratio analysis (SCIRA). According to this standard method, the direct addition of HFCS to honey would be detected when the adulteration was greater than 7%; this is the internationally recognized threshold level for samples to be considered adulterated [68,69].

### 3.3. Quantification of SBH Adulteration Level Using Different Regression Methods

Supervised quantification of the SBH adulteration level was studied using three different multivariate regression methods: PLSR, PCR, and MLR. For PLSR and PCR, fluorescence spectra from 357–725.5 nm (n = 738 wavelengths) were used as predictors (x-variables). For MLR, eight peaks were selected as x-variables based on their important fluorescence information: 378 nm, 384 nm, 391 nm, 470 nm, 477 nm, 490 nm, 540 nm, and the intensity ratio of 378 nm and 477 nm (I_378_/I_477_). The adulteration level (0–60%) was assigned as the target (y-variables). To minimize the risk of overfitting, the PLSR, PCR, and MLR were validated using an external validation sample set (*t*-test validation). The scatter plot of the developed calibration model and the validation are presented in Figure 8, Figure 9 and Figure 10 for PLSR, PCR, and MLR, respectively. The PLSR calibration model was acceptable, with R^2^ = 0.9637, RMSEC = 4.0792%, and SEC = 4.0907%. In the validation step, a good agreement between actual and predicted levels of adulteration was achieved with an R^2^ = 0.9610, RMSEV = 4.2418%, and SEV = 4.2608%. An even better result was obtained for PCR with lower error and higher R^2^ (RMSEC = 3.7014%, SEC = 3.7119%, R^2^ = 0.9701 in calibration, and RMSEV = 3.8347%, SEV = 3.8506%, R^2^ = 0.9681 for validation). Interestingly, the calibration and validation results of MLR with fewer x-variables are also acceptable, with an R^2^= 0.9480 for both calibration and validation. The error obtained in MLR is also acceptable (RMSEC = 4.8836%, SEC = 4.8974% for calibration and RMSEV = 4.9007%, SEV = 4.9206% for validation). It was noted that the RMSEC and RMSEV along with SEC and SEV were quite close (with a difference of less than 1%) for all regression models; indicating no over-fitting occurred. This study also showed that the PCR model with the highest R^2^ both in calibration and validation outperformed the PLSR and MLR models.

The quality of the regression models obtained in this study is comparable with previously reported studies. For example, Chen et al. [70] studied the application of three-dimensional fluorescence spectroscopy and PLSR to the authentication of four types of honey (sunflower, longan, buckwheat, and rape) adulterated by rice syrup. They obtained R^2^= 0.9495~0.9972. Ferreiro-Gonzalez et al. [71] demonstrated the success of visible-near-infrared spectroscopy applications coupled with PLSR for the prediction of honey adulteration with fructose-rich corn syrup. An R^2^ = 0.9990 and R^2^= 0.9855 were obtained for calibration and validation, respectively. UV-VIS (ultraviolet-visible) and NIR (near-infrared) spectroscopy were utilized to predict the fructose corn syrups in Acacia honey from Croatia (adulteration range: 10–90% *w*/*w*). Two regression models were developed for quantification of the adulteration level based on PLSR and MLP ANN (multiple layer perceptron neural networks). Using PLSR, R^2^ = 0.9268, and R^2^ = 0.9100 were reported for calibration and validation, respectively [26]. Possible quantification of glucose syrup adulteration in Acacia honey in the range of 10–90% (*w*/*w*) from Croatia was also investigated using NIR spectroscopy, resulting in the best PLSR model using pre-treated NIR spectra with R^2^ = 0.8978 for calibration and R^2^ = 0.8557 for validation [72]. Raypah et al. [73] studied SBH adulteration using visible-near infrared (VIS-NIR) spectroscopy combined with aquaphotomics. The direct adulteration was done by mixing high fructose syrup with the pure SBH samples in the range of adulteration from 10 to 90% (*v*/*v*). The VIS-NIR spectra from 400 to 1100 nm were pre-treated using smoothing and detrend (1st polynomial) and were used as x-variables, while the adulteration level (10–90% *v*/*v*) was input as a y-variable. The PLSR result was accurate, with R^2^ = 0.98 and RMSEC = 3.93% for calibration and R^2^ = 0.96 and RMSEV = 5.88% for validation. Visible and NIR spectroscopy was used to quantify different types of adulterants (inverted sugar, rice syrup, brown cane sugar, and fructose syrup) added to high-quality honey (Granada Protected Designation of Origin, Spain) at adulteration levels of 5–50% *w*/*w* [74]. Individual and global models based on PLSR were developed with R^2^ = 0.964~0.990 and RMSEC = 1.621~3.195%. Laser-induced breakdown spectroscopy (LIBS) along with four different PLSR methods was utilized to quantify the HFCS-55 and HFCS-90 in Acacia honey. R^2^ = 0.931~0.966 and RMSEC = 5.6~7.9% could be obtained [75]. In a recent study, NIR spectroscopy was combined with chemometrics to evaluate the adulteration in the SBH sample [76]. The adulterants were distilled water and apple cider vinegar. The PLSR model was developed with R^2^= 0.986049, bias= −0.010%, and RMSECV = 1.686% [76]. Raman spectroscopy was recently applied along with PLSR and ANN to quantify the adulteration level of a common single-flowered lychee honey adulterated with four syrups, HFCS, RS (rice syrup), MS (maltose syrup), and BS (blended syrup), in the range of adulteration 5–90% (*w*/*w*) [77]. The calibration model of PLSR with 6 LVs for HFCS quantification gave R^2^ = 0.9997, RMSEC = 0.6270% for calibration, and R^2^ = 0.9910, RMSECV = 3.2633% for validation. It is noted that a difference of 2.6363% was observed between RMSEC and RMSECV, which is bigger than that of our present study. Wu et al. [78] reported a similar result using Raman spectroscopy and PLSR for adulteration quantification of Acacia honey, litchi honey, and linden honey mixed with four syrups, including HFCS-55, RS, MS, and BS, with adulteration concentrations ranging from 5 to 60% (*w*/*w*). The quantification of multiple syrups in adulterated samples was performed using PLSR with acceptable results. The performance of our regression models was also comparable to the result of honey adulteration quantification using the standard method of HPLC and linear regression with R^2^ = 0.9835 [69].

The developed PLSR, PCR, and MLR models were used to predict the adulteration level in an independent prediction sample set (n = 52 samples). The slopes for the PLSR, PCR, and MLR models were 0.96, 0.95, and 0.94, respectively (Figure 11). Table 6 summarizes the performance of the prediction result. It is noted that all models resulted in satisfactory prediction results with the coefficient of determination in prediction (R^2^_p_) greater than 0.90 for the PLSR, PCR, and MLR models. The error was low both in terms of RMSEP, SEP, and bias. To compare our prediction performance with previously reported studies, two parameters, RER and RPD, are frequently used. According to Parrini et al. [79], the model can be considered sufficient for screening if the RPD is between 1.5 and 2.5. An excellent and acceptable prediction result must have an RER of more than 10 and an RPD of more than 2.5, as adopted by several studies [80,81,82]. All developed models are superior with acceptable RER and RPD values (RER = 13.16~14.81; RPD = 4.76~5.36), as seen in Table 6. Compared to previous studies on the quantification of honey adulteration using several different spectroscopy methods, our present results are better in terms of RER and RPD values. Nespeca et al. [83] used laser-induced breakdown spectroscopy (LIBS) for the detection and quantification of honey adulteration (range of HFCS and sugar cane syrup adulteration 5–95% *w*/*w*) and developed an acceptable PLSR model with RPD = 2.7. A report has been published on the use of visible and near-infrared (VIS-NIR) spectroscopy and PLSR with a *t*-test validation method for the detection of glucose concentration in a mixture of Saudi and imported honey samples (adulteration range: 0–33% *w*/*w*). An RPD of 2.06 was obtained [84]. A similar result was reported on the application of UV-VIS and NIR spectroscopy for quantification of fructose corn syrup in Acacia honey in the range of adulteration (10–90% *w*/*w*). The best PLSR model had RPD = 3.3150 and RER = 10.4512 [73]. A quantitative study was conducted using NIR spectroscopy and PLSR to predict the glucose syrup adulteration in Acacia honey (10–90% *w*/*w*). The best model was obtained for multiplicative scatter correction (MSC) spectral data with RPD = 2.7601 and RER = 8.7157 [74].

The LOD of the regression models varied between 12.79 and 14.55%. In recent work, honey adulteration using brown rice syrup, corn syrup, glucose syrup, sugar cane syrup, and wheat syrup (10–50% *w*/*w*) of Western Australian honey was quantified using benchtop ^1^H NMR (nuclear magnetic resonance) spectroscopy with a LOD = 5% *w*/*w* [85]. Our result was worse compared to the standard analytical methods of HPLC and SCIRA for honey adulteration. Ultrahigh-performance liquid chromatography with quadrupole time-of-flight mass spectrometry (UHPLC-Q-TOF-MS) successfully determined all the honey adulterants simultaneously with a detection range above 10% [86]. SCIRA methods can detect only up to 7% (*w*/*w*), whereas HPLC was found to detect as low as 2.5% (*w*/*w*) adulterants [86]. The LOD and LOQ in this present study were higher compared to several previous studies due to the high range of adulterated SBH samples (10–60% *v*/*v*) used in the study. However, as demonstrated in Figure 11, the prediction plot, most predicted samples were very close to the regression line, which indicates that the developed PLSR, PCR, and MLR models could be used to detect HFCS adulteration of pure SBH samples.

## 4. Conclusions

This study reported for the first time the application of portable LED-based fluorescence spectroscopy for non-targeted detection and quantification of SBH samples adulterated with HFCS-55. It was clear that the typical fluorescence spectral data of pure and adulterated SBH samples was similar in shape but different in intensity. According to this result, a further qualitative study was conducted to classify and discriminate between pure and adulterated SBH samples. The results of PCA and SIMCA showed a possible separation between pure and adulterated SBH samples, with accuracy, sensitivity, specificity, and precision reaching 100%. Several important wavelengths with high x-loadings were identified based on SIMCA results, including wavelengths at 378, 384, 391, 470, 477, 490, and 540 nm and the intensity ratio of 378 nm and 477 nm (I_378_/I_477_). Using these important wavelengths, a quantification of adulteration level based on MLR was successfully developed with acceptable performance, where RPD = 4.76 and RER = 13.16. In conclusion, this current research successfully provides new insights regarding the application of portable LED-based fluorescence spectroscopy for the identification of SBH. The proposed portable LED-based fluorescence spectroscopy can be applied to measure spectral information in various locations, including field environments, remote sites, and non-laboratory settings. In addition, the methods could work with very small sample volumes, making them suitable for applications where sample availability is limited or precious, such as SBH. Additionally, even though the statistical model used is simple and common, it gives high accuracy in the discrimination of pure and adulterated SBH samples. However, the sample number in this study is limited. To generalize the results and provide a more robust model, measurement of various samples of SBH with different botanical, entomological, and geographical origins is required. This portable fluorescence spectrometer could be connected to a smartphone via Bluetooth and easily connect to the internet to develop an IoT for honey authentication in the field. In turn, it could be used to support rapid, accurate, and affordable food traceability.

## Figures and Tables

**Figure 1 foods-12-03067-f001:**
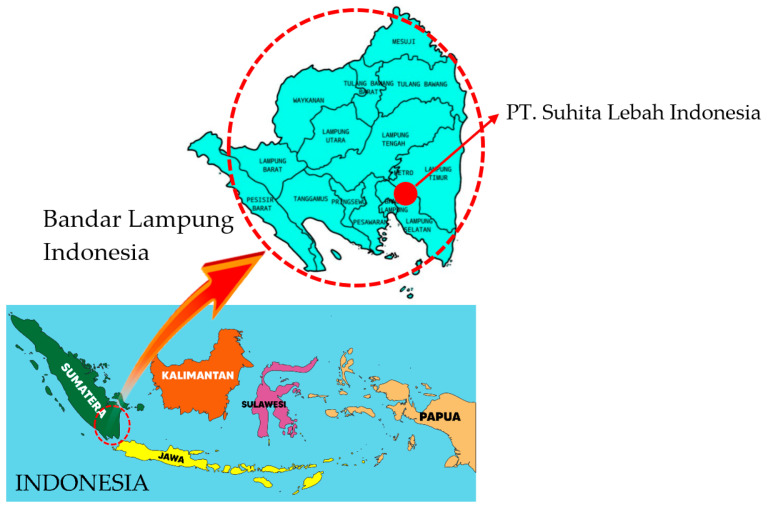
Geographical origin of SBH *Heterotrigona itama* in Bandar Lampung, Sumatra, Indonesia.

**Figure 2 foods-12-03067-f002:**
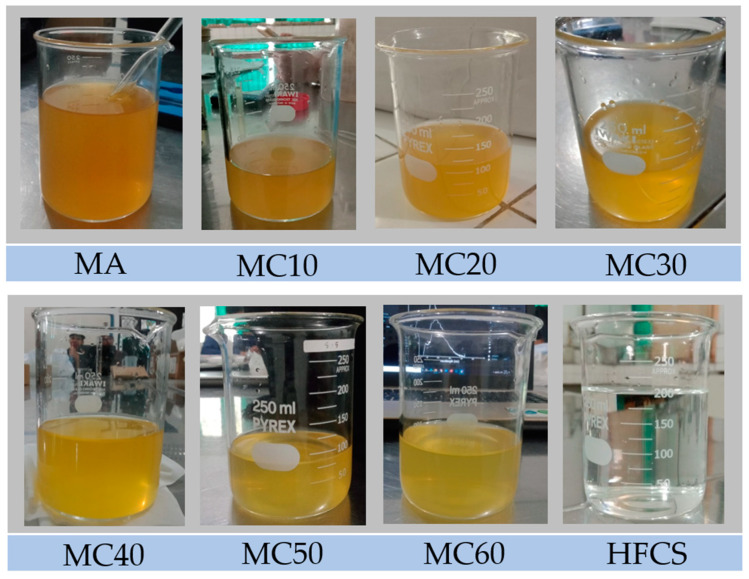
The visual appearance of SBH *Heterotrigona itama* honey and their adulterated samples.

**Figure 3 foods-12-03067-f003:**
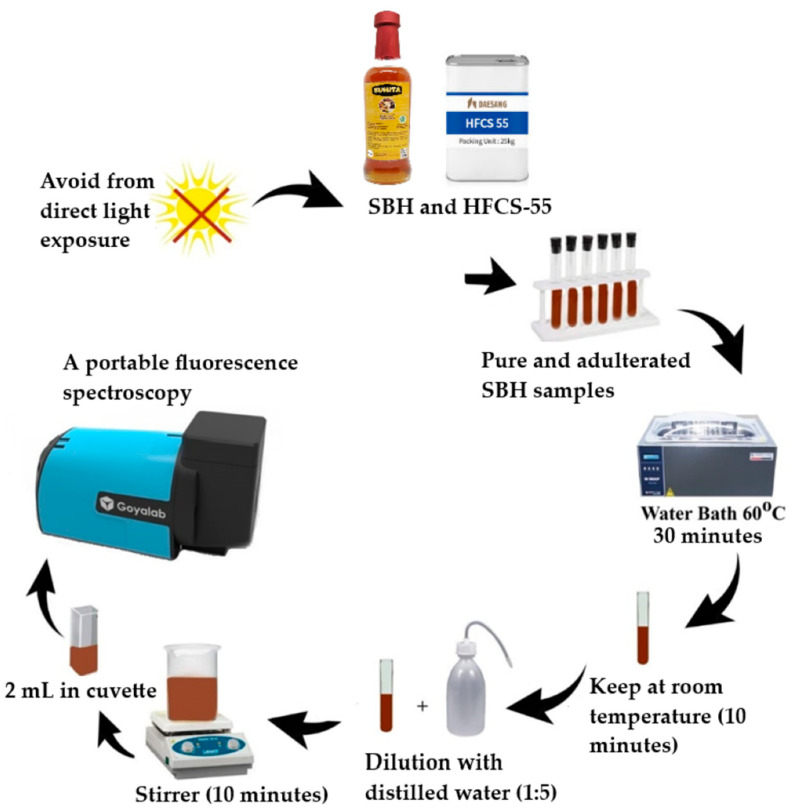
Sample preparation protocol and fluorescence spectral data acquisition of pure and adulterated SBH samples.

**Figure 4 foods-12-03067-f004:**
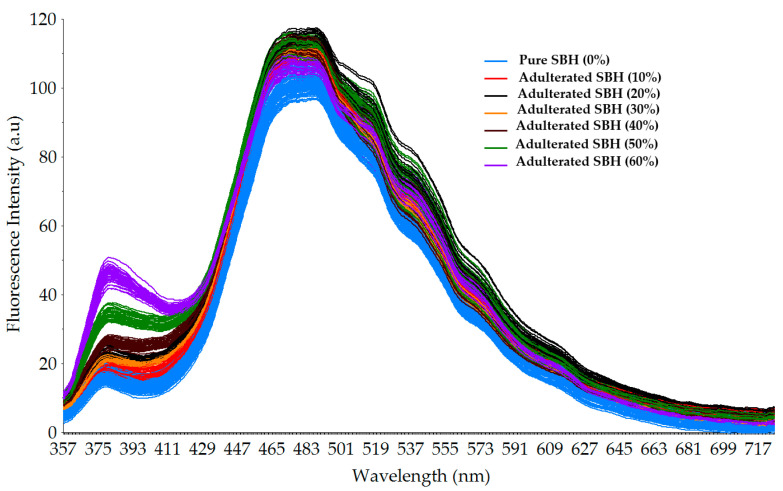
Smoothed fluorescence spectral data of pure and adulterated SBH samples in the range of 357–725.5 nm was obtained using 4 LED lamps as excitation sources (λ = 365 nm).

**Figure 5 foods-12-03067-f005:**
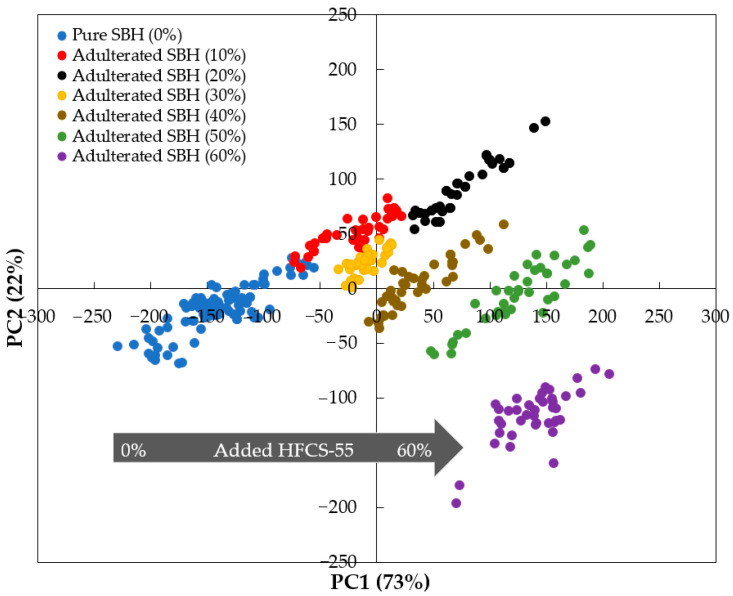
Unsupervised PCA score plots for the smoothed fluorescence spectral data collected for pure and adulterated SBH samples in the range of 357–725.5 nm.

**Figure 6 foods-12-03067-f006:**
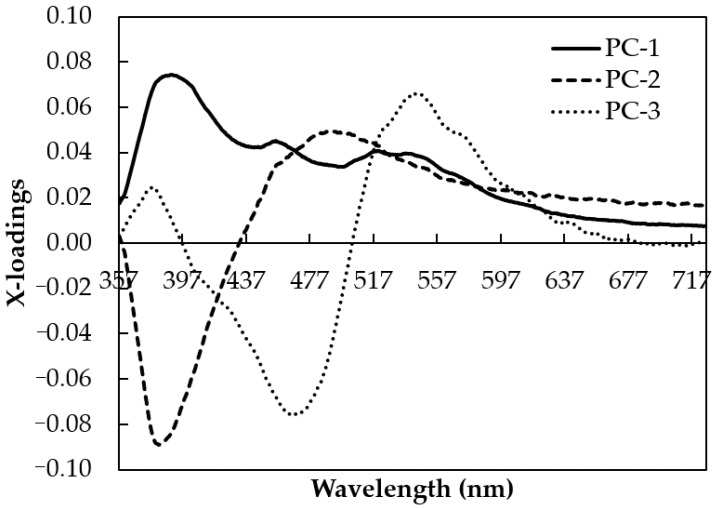
Unsupervised PCA x-loading plots for the smoothed fluorescence spectral data collected for pure and adulterated SBH samples in the range of 357–725.5 nm.

**Figure 7 foods-12-03067-f007:**
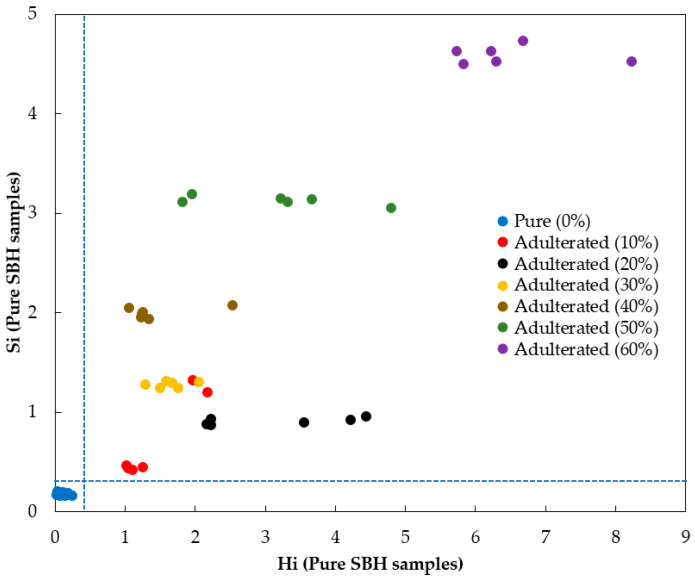
The plot of sample-to-model distance (Si) versus sample leverage (Hi) of pure SBH samples.

**Figure 8 foods-12-03067-f008:**
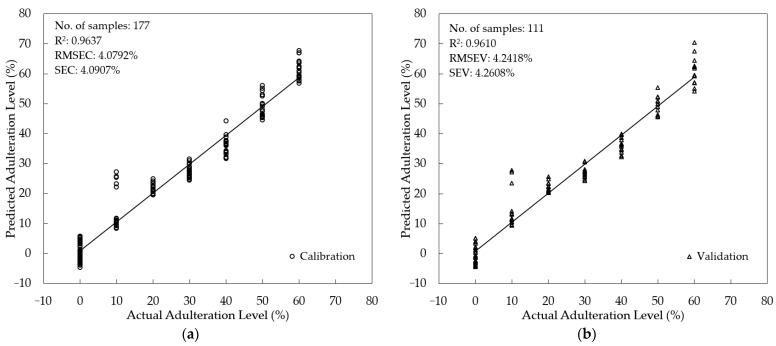
Model development for the quantification of the adulteration level using PLSR (**a**) Calibration; (**b**) validation.

**Figure 9 foods-12-03067-f009:**
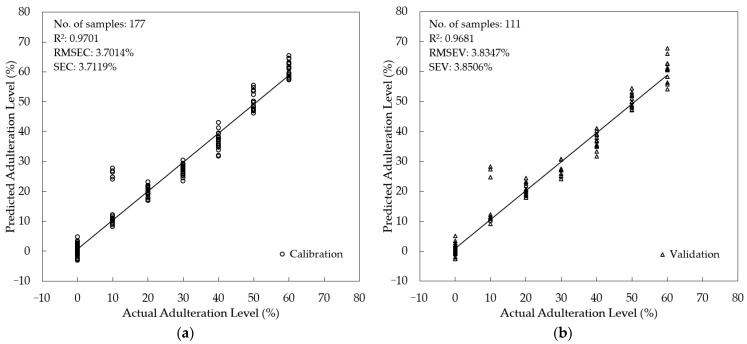
Model development for the quantification of the adulteration level using PCR (**a**) Calibration; (**b**) validation.

**Figure 10 foods-12-03067-f010:**
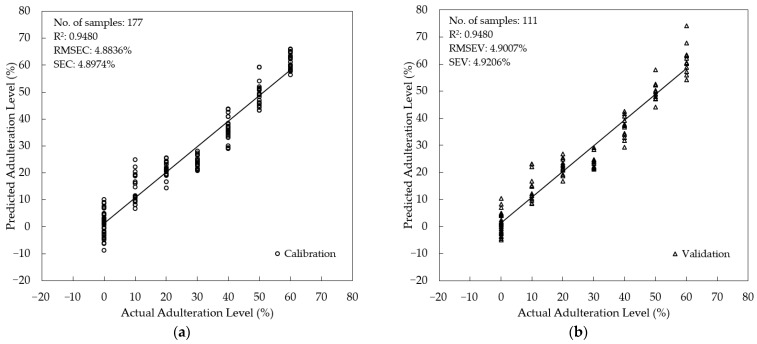
Model development for the quantification of the adulteration level using MLR (**a**) Calibration; (**b**) validation.

**Figure 11 foods-12-03067-f011:**
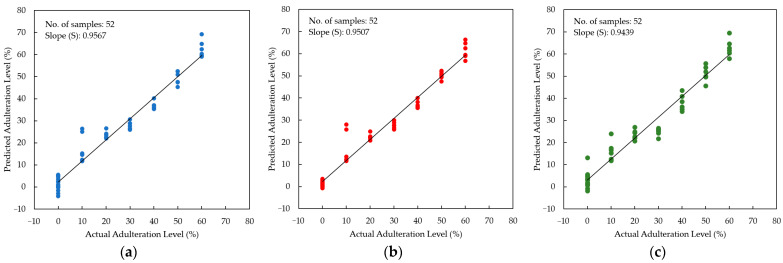
Scatter plot between actual and predicted adulteration levels in the prediction step (**a**) PLSR, (**b**) PCR, and (**c**) MLR.

**Table 1 foods-12-03067-t001:** Detailed information on pure and adulterated SBH samples used in this study.

Sample Code	Adulteration Level (%) ^1^	Number of Samples
MA	0	100
MC10	10	40
MC20	20	40
MC30	30	40
MC40	40	40
MC50	50	40
MC60	60	40

^1^ adulteration level was expressed in % (*v*/*v*).

**Table 2 foods-12-03067-t002:** Sample distribution in calibration, validation, and prediction sets of pure and adulterated SBH classes for SIMCA calculation.

Classes	Calibration Set	Validation Set	Prediction Set
Pure SBH	51	33	16
Adulterated SBH	126	78	36

**Table 3 foods-12-03067-t003:** Calibration, validation, and prediction sets of pure and adulterated SBH samples were used in this study.

Items	Calibration Set	Validation Set	Prediction Set
Number of samples	177	111	52
Range ^1^	0–60	0–60	0–60
Standard deviation (SD) ^1^	21.48	21.57	21.72
Mean ^1^	24.92	24.59	24.23

^1^ The range, mean, and standard deviations were expressed in % (*v*/*v*).

**Table 4 foods-12-03067-t004:** The result of SIMCA model development for pure and adulterated SBH classes.

Principal Components (PCs)	Cumulative Percent Variance (CPV) (%)			
	Pure SBH		Adulterated SBH	
	Calibration	Validation	Calibration	Validation
PC1	84.08973	80.41664	63.44584	63.91681
PC2	94.11534	91.30837	93.14684	93.60857
PC3	98.08557	96.70327	98.38311	98.59088
PC4	99.64038	99.54868	99.70518	99.66491
PC5	99.73003	99.60417	99.88976	99.85424
PC6	99.77257	99.64778	99.90607	99.86829

**Table 5 foods-12-03067-t005:** Confusion matrix based on using the developed SIMCA model for classification of pure and adulterated SBH classes.

		Predicted Class		
		Pure SBH	Adulterated SBH	Total
Actual Class	Pure SBH	True Positive (TP) = 16	False Negative (FN) = 0	16
	Adulterated SBH	False Positive (FP) = 0	True Negative (TN) = 34	34
	Total	16	34	

**Table 6 foods-12-03067-t006:** The result of prediction using three different regression methods for quantification of the adulteration level.

Regressions	R^2^_p_	RMSEP ^1^	SEP ^1^	Bias ^1^	RER	RPD	LOD ^1^	LOQ ^1^
PLSR	0.9566	4.4818	4.3547	1.2197	13.79	4.99	13.59	45.31
PCR	0.9627	4.1579	4.0469	1.1073	14.81	5.36	12.79	42.63
MLR	0.9497	4.8259	4.5601	1.7015	13.16	4.76	14.55	48.51

^1^ RMSEP, SEP, bias, LOD, and LOQ were expressed in % (*v*/*v*).

## Data Availability

The datasets generated for this study are available upon reasonable request from the corresponding authors.

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
