# Peer review of "Non-Targeted Detection and Quantification of Food Adulteration of High-Quality Stingless Bee Honey (SBH) via a Portable LED-Based Fluorescence Spectroscopy"

_foods, 2023, doi:10.3390/foods12163067_

Round 1
Reviewer 1 Report
The manuscript is written with clear understanding of the project addressed, but there are some problems. My specific comments are as follows:
In line 77-79, this sentence should be based on references. What is the several advantages? The several advantages must be specific.
The pure and adulterated prediction samples were classified with 100% accuracy, sensitivity, specificity and precision, had an excellent classification. However, as shown in figure 2, the pure and adulterated samples can be classified by the visual appearance. The color difference between pure samples and adulterated samples is obvious. Therefore, the manuscript has little research significance.
Author Response
Dear Reviewer,
Thank you for the valuable comments and suggestions. We really appreciate it. Herewith we sent the replies. Please find it in the attachment.

Reviewer 2 Report
Comments to Authors:
1. Abstract: Add some details of results in the abstract rather than focusing much on PCA.
2. Introduction: The introduction section looks good, however authors still need to add details on limitations of LED based methods and applications in other segments.
Material and methods:
1. Line 95-98: provide GPS location for the sample collection.
2. It will be nice to illustrate the chemometric analysis to get fair idea for readers having less statistical understandings.
Results and discussions:
1. Is the current research set the landmark analysis of SBH? How it could be generalise. What authors recommend according to the lambda max and intensity?
2. How current methods should be accurate to detect minute like 1-5 % adulteration.
3. I appreciate authors for proper justification of finding with statistical modelling.
4. Provide the strengths and limitations of the proposed analysis method and statistical modelling.
Need to improve.
Author Response
Dear Reviewer,
Thank you very much for your valuable comments and suggestions. Herewith we sent you the replies. Please find it in the attachment file.
Thank you

Round 2
Reviewer 2 Report
Authors appropriately addressed the concerns raised by me, thus I hereby recommend to accept the said manuscript
Acceptable